# DISTRIBUTION MATCHING PROTOTYPICAL NETWORK FOR UNSUPERVISED DOMAIN ADAPTATION

## ABSTRACT

State-of-the-art Unsupervised Domain Adaptation (UDA) methods learn transferable features by minimizing the feature distribution discrepancy between the source and target domains. Different from these methods which do not model the feature distributions explicitly, in this paper, we explore explicit feature distribution modeling for UDA. In particular, we propose Distribution Matching Prototypical Network (DMPN) to model the deep features from each domain as Gaussian mixture distributions. With explicit feature distribution modeling, we can easily measure the discrepancy between the two domains. In DMPN, we propose two new domain discrepancy losses with probabilistic interpretations. The first one minimizes the distances between the corresponding Gaussian component means of the source and target data. The second one minimizes the pseudo negative log likelihood of generating the target features from source feature distribution. To learn both discriminative and domain invariant features, DMPN is trained by minimizing the classification loss on the labeled source data and the domain discrepancy losses together. Extensive experiments are conducted over two UDA tasks. Our approach yields a large margin in the Digits Image transfer task over state-of-the-art approaches. More remarkably, DMPN obtains a mean accuracy of 81.4% on VisDA 2017 dataset. The hyper-parameter sensitivity analysis shows that our approach is robust w.r.t hyper-parameter changes.

## 1 INTRODUCTION

Recent advances in deep learning have significantly improved state-of-the-art performance for a wide range of applications. However, the improvement comes with the requirement of a massive amount of labeled data for each task domain to supervise the deep model. Since manual labeling is expensive and time-consuming, it is therefore desirable to leverage or reuse rich labeled data from a related domain. This process is called domain adaptation, which transfers knowledge from a label rich source domain to a label scarce target domain (Pan & Yang, 2009).

Domain adaptation is an important research problem with diverse applications in machine learning, computer vision (Gong et al., 2012; Gopalan et al., 2011; Hoffman et al., 2014; Saenko et al., 2010) and natural language processing (Collobert et al., 2011; Glorot et al., 2011). Traditional methods try to solve this problem via learning domain invariant features by minimizing certain distance metric measuring the domain discrepancy, for example Maximum Mean Discrepancy (MMD) (Gretton et al., 2009; Pan et al., 2008; 2010) and correlation distance (Sun & Saenko, 2016). Then labeled source data is used to learn a model for the target domain. Recent studies have shown that deep neural networks can learn more transferable features for domain adaptation (Glorot et al., 2011; Yosinski et al., 2014). Consequently, adaptation layers have been embedded in the pipeline of deep feature learning to learn concurrently from the source domain supervision and some specially designed domain discrepancy losses (Tzeng et al., 2014; Long et al., 2015; Sun & Saenko, 2016; Zellinger et al., 2017).

However, none of these methods explicitly model the feature distributions of the source and target data to measure the discrepancy. Inspired from the recent works by Wan et al. (2018) and Yang et al. (2018), which have shown that modeling feature distribution of a training set improves classification performance, we explore explicit distribution modeling for UDA. We model the feature distributions

as Gaussin mixture distributions, which facilitates us to measure the discrepancy between the source and target domains.

Our proposed method, i.e., DMPN, works as follows. We train a deep network over the source domain data to generate features following a Gaussian mixture distribution. The network is then used to assign pseudo labels to the unlabeled target data. To learn both discriminative and domain invariant features, we fine-tune the network to minimize the cross-entropy loss on the labeled source data and domain discrepancy losses. Specifically, we propose two new domain discrepancy losses by exploiting the explicit Gaussian mixture distributions of the deep features. The first one minimizes the distances between the corresponding Gaussian component means between the source and target data. We call it Gaussian Component Mean Matching (GCMM). The second one minimizes the negative log likelihood of generating the target features from the source feature distribution. We call it Pseudo Distribution Matching (PDM). Extensive experiments on Digits Image transfer tasks and synthetic-to-real image transfer task demonstrate our approach can provide superior results than state-of-the-art approaches. We present our proposed method in Section 3, extensive experiment results and analysis in Section 4 and conclusion in Section 5.

## 2 RELATED WORKS

**Domain adaptation** is an important research problem with diverse applications in machine learning, computer vision (Gong et al., 2012; Gopalan et al., 2011; Hoffman et al., 2014; Saenko et al., 2010) and natural language processing (Collobert et al., 2011; Glorot et al., 2011). According to the survey Pan & Yang (2009), traditional domain adaptation methods can be organized into two categories: feature matching and instance re-weighting. Feature matching aims to reduce the domain discrepancy via learning domain invariant features by minimizing certain distance metric, for example Maximum Mean Discrepancy (MMD) (Gretton et al., 2009; Pan et al., 2008; 2010), correlation distance (Sun & Saenko, 2016), Central Moment Discrepancy (CMD) Zellinger et al. (2017) and et al. Then labeled source data is used to learn a model for the target domain. Instance re-weighting aims to reduce the domain discrepancy via re-weighting the source instances according to their importance weights with respect to the target distribution (Huang et al., 2007).

In the era of deep learning, studies have shown that deep neural networks can learn more transferable features for domain adaptation (Glorot et al., 2011; Yosinski et al., 2014), therefore, domain adaptation layers have been embedded in the pipeline of deep feature learning to learn concurrently from the source domain supervision and some specially designed domain discrepancy losses (Tzeng et al., 2014; Long et al., 2015; Sun & Saenko, 2016; Zellinger et al., 2017). Some recent works Ganin & Lempitsky (2014), Tzeng et al. (2017), Long et al. (2018) add a domain discriminator into the deep feature learning pipeline, where a feature generator and a domain discriminator are learned adversarially to generate domain invariant features. All these works can be categorized as the feature matching type of domain adaptation method. However, none of them models the feature distributions of the source and target data for distribution matching. In this paper, we show that explicitly modeling the feature distributions enables us to measure the domain discrepancy more easily and helps us to propose new domain discrepancy losses.

**Prototypical network** (PN) was first proposed in Snell et al. (2017) for few shot learning, which shows that learning PN is equivalent to performing mixture density estimation on the deep features with an exponential density. Recently, in Wan et al. (2018)'s and Yang et al. (2018)'s works, it has been shown that modeling the deep feature distribution of a training set as Gaussian mixture distribution improves classification performance. As Gaussian density belongs to one type of exponential density, the models proposed in Wan et al. (2018)'s and Yang et al. (2018)'s works are variants of PN. However, the two works study the classification problem in a single domain, which is different from our work on the problem of domain adaptation. In Pan et al. (2019), prototypical networks are first applied for domain adaptation. Multi-granular domain discrepancy minimization at both class-level and sample-level are employed in Pan et al. (2019) to reduce the domain difference and achieves state-of-the-art results in various domain adaptation tasks. However, in Pan et al. (2019)'s work, the deep feature distribution is modeled implicitly when they apply PN for UDA, in our work, we explicitly model the deep feature distribution as Gaussian mixture distribution for UDA.

## 3 Distribution Matching Prototypical Networks

In Unsupervised Domain Adaptation (UDA), we are given $N^s$ labeled samples $\mathbb{D}^s = \{(\boldsymbol{x}_i^s, y_i^s)\}_{i=1}^{N^s}$ in the source domain and $N^t$ unlabeled samples $\mathbb{D}^t = \{\boldsymbol{x}_i^t\}_{i=1}^{N^t}$ in the target domain. The source and target samples share the same set of labels and are sampled from probability distributions $P^s$ and $P^t$ respectively with $P^s \neq P^t$. The goal is to transfer knowledge learnt from the labeled source domain to the unlabeled target domain.

### 3.1 Deep Feature Distribution Modeling

We model the deep embedded features of the source data as a Gaussian mixture distribution where the Gaussian component means act as the prototypes for each class. Let $\{\boldsymbol{\mu}_c^s\}_{c=1}^C$ and $\{\boldsymbol{\Sigma}_c^s\}_{c=1}^C$ be the Gaussian component means and covariance matrices of the Gaussian mixture distribution, then the posterior distribution of a class $y$ given the embedded feature $\boldsymbol{f}$ can be expressed as in Eqn. 1 where $\boldsymbol{f} = F(\boldsymbol{x}, \boldsymbol{\theta})$, $F : \mathbb{X} \to \mathbb{R}^d$ is the embedding function with parameter $\boldsymbol{\theta}$ and $d$ is the dimension of the embedded feature, $p(c)$ is the prior probability of class $c$ and $C$ is the total number of classes.

$$p(y|f) = \frac{\mathcal{N}(\boldsymbol{f}; \boldsymbol{\mu}_y^s, \boldsymbol{\Sigma}_y^s)p(y)}{\sum_{c=1}^C \mathcal{N}(\boldsymbol{f}; \boldsymbol{\mu}_c^s, \boldsymbol{\Sigma}_c^s)p(c)} \tag{1}$$

With labeled source data $\mathbb{D}^s = \{(\boldsymbol{x}_i^s, y_i^s)\}_{i=1}^{N^s}$, a classification loss $\mathcal{L}_{cls}$ can be computed as the cross-entropy between the posterior probability distribution and the one-hot class label as shown in Eq. 2 and following Wan et al. (2018), a log likelihood regularization term $\mathcal{L}_{lkd}$ can be defined as in Eq. 3, where $\boldsymbol{f}_i^s = F(\boldsymbol{x}_i^s, \boldsymbol{\theta})$.

$$\mathcal{L}_{cls}(\mathbb{D}^s) = -\frac{1}{N^s}\sum_{i=1}^{N^s} log \frac{\mathcal{N}(\boldsymbol{f}_i^s; \boldsymbol{\mu}_{y_i^s}^s, \boldsymbol{\Sigma}_{y_i^s}^s)p(y_i^s)}{\sum_{c=1}^C \mathcal{N}(\boldsymbol{f}_i^s; \boldsymbol{\mu}_c^s, \boldsymbol{\Sigma}_c^s)p(c)} \tag{2}$$

$$\mathcal{L}_{lkd}(\mathbb{D}^s) = -\frac{1}{N^s}\sum_{i=1}^{N^s} log \mathcal{N}(\boldsymbol{f}_i^s; \boldsymbol{\mu}_{y_i^s}^s, \boldsymbol{\Sigma}_{y_i^s}^s) \tag{3}$$

The final loss function $\mathcal{L}_{GM}$ for training a network with Gaussian mixture feature distribution is defined as $\mathcal{L}_{GM} = \mathcal{L}_{cls}(\mathbb{D}) + \varphi \mathcal{L}_{lkd}(\mathbb{D})$, where $\varphi$ is a non-negative weighting coefficient. Notice, the distribution parameters $\{\boldsymbol{\mu}_c^s\}_{c=1}^C$ and $\{\boldsymbol{\Sigma}_c^s\}_{c=1}^C$ are learned automatically from data.

### 3.2 Gaussian Component Mean Matching

To match the deep feature distributions between the source and target data, we propose to match the corresponding Gaussian component means between them. We utilize the network learnt on the labeled source data to assign pseudo labels to target samples. As such, we denote the target samples with pseudo labels as $\hat{\mathbb{D}}^t = \{(\boldsymbol{x}_i^t, \hat{y}_i^t)\}_{i=1}^{N^t}$. We empirically estimate the Gaussian component means $\{\boldsymbol{\mu}_c^{es}\}_{c=1}^C$ [1] and $\{\boldsymbol{\mu}_c^{et}\}_{c=1}^C$ as follows:

$$\boldsymbol{\mu}_c^{es} = \frac{1}{|\mathbb{D}_c^s|}\sum_{\boldsymbol{x}_i^s \in \mathbb{D}_c^s} \boldsymbol{f}_i^s, \qquad \boldsymbol{\mu}_c^{et} = \frac{1}{|\mathbb{D}_c^t|}\sum_{\boldsymbol{x}_i^t \in \mathbb{D}_c^t} \boldsymbol{f}_i^t \tag{4}$$

where $\mathbb{D}_c^s$ and $\hat{\mathbb{D}}_c^t$ denote the sets of source/target samples from class $c$, $\boldsymbol{f}_i^t = F(\boldsymbol{x}_i^t; \boldsymbol{\theta})$. The Gaussian Component Mean Matching (GCMM) loss $\mathcal{L}_{GCMM}$ is defined as follows:

$$\mathcal{L}_{GCMM}(\{\boldsymbol{\mu}_c^{es}\}_{c=1}^C, \{\boldsymbol{\mu}_c^{et}\}_{c=1}^C) = \frac{1}{C}\sum_{c=1}^C ||\boldsymbol{\mu}_c^{es} - \boldsymbol{\mu}_c^{et}||^2 \tag{5}$$

where $|| \cdot ||$ is the $L^2$ norm between two vectors. Intuitively, if the source features and target features follow the same Gaussian mixture distribution, then the Gaussian component means of the same class from the two domains will be the same. Thus minimizing $\mathcal{L}_{GCMM}$ helps to reduce the domain discrepancy. Better illustrated in Fig. 1

---

[1] $\{\boldsymbol{\mu}_c^{es}\}_{c=1}^C$ is different from $\{\boldsymbol{\mu}_c^s\}_{c=1}^C$, as the latter are learned directly from data and are used to assign pseudo labels for target data.

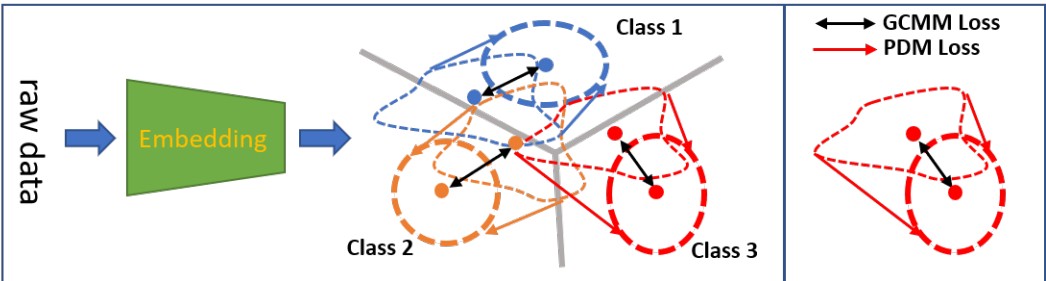

Figure 1: Illustration of the overall training objective. This figure displays the model after we finish pre-training it with the labeled source data on $\mathcal{L}_{GM}$. Different colors represent different classes. Dotted ellipses represent Gaussian mixture distribution of the source embedded features. The amorphous shapes represent pseudo labeled target feature distribution before we optimize the network further on the overall objective function in Eqn. 7. GCMM loss tries to bring the corresponding Gaussian component means between the source data and pseudo labeled target data closer, represented by the black two-way arrows. Minimizing GCMM brings the feature distributions of the source and target domains closer, thus reducing the domain discrepancy. PDM loss tries to match the pseudo target feature distribution to the source Gaussian mixture distribution, represented by the colored one-way arrow. Minimizing PDM increases the likelihood of target features on the source feature distribution, thus reducing the domain discrepancy. Best viewed in color.

### 3.3 PSEUDO DISTRIBUTION MATCHING

On the pseudo labeled target data $\hat{\mathbb{D}}^t$, we further propose to match the embedded target feature distribution with the source Gaussian mixture feature distribution via minimizing the following pseudo negative log likelihood loss, which we denoted as $\mathcal{L}_{PDM}$:

$$\mathcal{L}_{PDM}(\hat{\mathbb{D}}^t) = -\frac{1}{|\hat{\mathbb{D}}^t|} \sum_{(x_i^t, \hat{y}_i^t) \in \hat{\mathbb{D}}^t} log\,\mathcal{N}(\boldsymbol{f}_i^t; \boldsymbol{\mu}_{\hat{y}_i}^s, \boldsymbol{\Sigma}_{\hat{y}_i}^s) \qquad (6)$$

Minimizing $\mathcal{L}_{PDM}{}^2$ maximizes the likelihood of the pseudo labeled target features on the source Gaussian mixture feature distribution. To achieve that, the network is enforced to learn an embedding function which produces similar embedded feature distributions between the source data and target data. Otherwise, this term will induce a large loss value and dominate the overall objective function to be minimized. Therefore, minimizing $\mathcal{L}_{PDM}$ helps to reduce the domain discrepancy. As we are using pseudo labeled target data to calculate this domain discrepancy loss function, we term it as Pseudo Distribution Matching (PDM) loss. Furthermore, while minimizing GCMM loss brings the source and target feature distribution closer, minimizing PDM loss shapes the target feature distribution to be similar as the source Gaussian mixture distribution. Thus, these two loss functions complement each other to reduce the distribution discrepancy. Better illustrated in Fig. 1.

### 3.4 OPTIMIZATION

The overall training objective of DMPN can be written as follows:

$$\min_{\theta, \{\boldsymbol{\mu}_c^s\}_{c=1}^C, \{\boldsymbol{\Sigma}_c^s\}_{c=1}^C} \mathcal{L}_{cls}(\mathbb{D}^s) + \varphi\mathcal{L}_{lkd}(\mathbb{D}^s) + \alpha\mathcal{L}_{GCMM}(\{\boldsymbol{\mu}_c^{es}\}_{c=1}^C, \{\boldsymbol{\mu}_c^{et}\}_{c=1}^C) + \beta\mathcal{L}_{PDM}(\hat{\mathbb{D}}^t) \quad (7)$$

where minimizing the first two terms of the objective function helps the model to learn discriminative features with the supervision from the labeled source data, and minimizing the last two terms helps to match the embedded feature distributions between the source and target domains so that the learned classifier from the labeled source data can be directly applied in the target domain. The whole model is illustrated in Fig. 1.

**Training Procedure.** To train DMPN, we first pre-train a network with labeled source data on $\mathcal{L}_{GM}$. Then mini-batch gradient descent algorithm is adopted for further optimization of the network on

---

[2]Notice, gradient from $\mathcal{L}_{PDM}$ does not back-propagate to update $\{\boldsymbol{\mu}_c^s\}_{c=1}^C$ and $\{\boldsymbol{\Sigma}_c^s\}_{c=1}^C$. We learn source distribution parameters only from labeled source data.

Eqn. 7, where half of the samples in the mini-batch are from labeled source data $\mathbb{D}^s$ and the other half are from unlabeled target data $\mathbb{D}^t$. To obtain pseudo labels for the unlabeled target data, we use the learned source distribution parameters to calculate the class probabilities for each target data point as in Eqn. 1 and assign the class with the largest probability as the pseudo label. To remedy the error of the self-labeling, we took similar approach as in French et al. (2018) and Pan et al. (2019) to filter unlabeled target data points whose maximum predicted class probability is smaller than some threshold. Apart from that, we also propose to weight the contribution of each sample to the discrepancy loss based on the predicted probability. In this way, less confidently predicted target samples will make smaller contributions in the training process.

**Inference.** For inference, we first apply the learned embedding function $F$ on the target data, then we will use the learned distribution parameters to calculate the class probabilities for each target data point as in Eqn. 1. Finally, we output the class with the largest probability for each target data point as our prediction.

## 3.5 THEORETICAL INSIGHT

There is another type of domain adaptation problem, called Supervised Domain Adaptation (SDA) in the literature. In SDA, we are provided with a large amount of labeled source data and a small amount of labeled target data, the goal is to find a hypothesis that works well in the target domain. By employing pseudo labeled target data in the training process, our method can be considered as working on a generalized problem of SDA, where the labeled target data is noisy. Ben-David et al. (2010) has proved that we can bound the target error of a domain adaptation algorithm that minimizes a convex combination of empirical source and target error in SDA as follows:

$$|\epsilon_\gamma(h) - \epsilon_t(h)| \leq (1 - \gamma)(\frac{1}{2}d_{\mathcal{H}\Delta\mathcal{H}}(P^s, P^t) + \lambda) \tag{8}$$

where $\epsilon_\gamma(h) = \gamma\epsilon_t(h) + (1 - \gamma)\epsilon_s(h)$ is the convex combination of the source and target error with $\gamma \in [0, 1]$, $\epsilon_s = \mathbb{E}_{x \sim P^s}[|h(x) - f^s(x)|]$, $\epsilon_t = \mathbb{E}_{x \sim P^t}[|h(x) - f^t(x)|]$, $f^s$ and $f^t$ are the labeling function in the source and target domains respectively, $h$ is a hypothesis in class $\mathcal{H}$, $d_{\mathcal{H}\Delta\mathcal{H}}(P^s, P^t) = 2\sup_{h,h' \in \mathcal{H}} |P_{x \sim P^s}[h(x) \neq h'(x)] - P_{x \sim P^t}[h(x) \neq h'(x)]|$ measures the domain discrepsancy in the hypothesis space $\mathcal{H}$ and $\lambda = \epsilon_s(h^*) + \epsilon_t(h^*)$ is the combined error in two domains of the joint ideal hypothesis $h^* = \arg\min_{h \in \mathcal{H}} \epsilon_s(h) + \epsilon_t(h)$.

Denote the noise ratio of the target labeling function to be $\rho$, the convex combination of the source and noisy target error as $\tilde{\epsilon}_\gamma(h) = \gamma\epsilon_{t'}(h) + (1 - \gamma)\epsilon_s(h)$, where $\epsilon_{t'}(h)$ is the target error on the noisy target labeling function, then we can bound the target error as follows:

$$
\begin{aligned}
|\tilde{\epsilon}_\gamma(h) - \epsilon_t(h)| &= |\gamma\epsilon_{t'}(h) + (1 - \gamma)\epsilon_s(h) - \epsilon_t(h) + \gamma\epsilon_t(h) - \gamma\epsilon_t(h)| \\
&\leq |\epsilon_\gamma(h) - \epsilon_t(h)| + \gamma|\epsilon_{t'}(h) - \epsilon_t(h)| \\
&\leq (1 - \gamma)(\frac{1}{2}d_{\mathcal{H}\Delta\mathcal{H}}(P^s, P^t) + \lambda) + \rho\gamma
\end{aligned}
\tag{9}
$$

In summary, this bound is decomposed into three parts: the domain discrepancy $d_{\mathcal{H}\Delta\mathcal{H}}$, the error $\lambda$ of the ideal joint hypothesis and the noise ratio $\rho$ of the pseudo labels. In DMPN, we minimize the first term through minimizing the domain discrepancy losses, as $d_{\mathcal{H}\Delta\mathcal{H}}$ is small when the source features and target features have similar distribution and minimizing the domain discrepancy losses makes the source and target feature to distribute similarly. The second term is assumed to be small, as otherwise there is no classifier that performs well on both domains. Finally, during training, as we continuously improve the accuracy of the classifier for target data, we get more and more accurate predictions, thus reducing the noise ratio $\rho$. We empirically verify that $\rho$ is decreasing in Section 4.2.

## 4 EXPERIMENTS

### 4.1 DATASETS AND EXPERIMENTAL SETTINGS

**Digits Image Transfer.** For the Digits Image transfer task, we consider the MNIST (M), USPS (U) and SVHN (S) datasets. Both the MNIST (M) and USPS (U) image datasets contains handwritten digits from '0' to '9'. The MNIST dataset consists of 70k images and the USPS dataset has 9.3k

images. Unlike MNIST and USPS, the SVHN (S) dataset is a real-world Digits dataset of house numbers in Google street view images and contains 100k cropped Digits images. We follow the standard evaluation protocol (Tzeng et al., 2017; Pan et al., 2019). We consider three directions of adaptation: M → U, U → M and S → M. For the transfer between MNIST and USPS, we sample 2k images from MNIST training set (60,000) and 1.8k images from USPS training set (7,291) for adaptation and evaluation is reported on the standard test sets: MNIST (10,000), USPS (2,007). For S → M, we use the whole training set SVHN (73,257) and MNIST (60,000) for adaptation and evaluation is reported on the standard test set MNIST (10,000). In addition, we use the same CNN architecture, namely a simple modified version of LeCun et al. (1998) (2 conv-layer LeNet) as Pan et al. (2019) and Tzeng et al. (2017) for fair comparison[3].

**Synthetic-to-Real Image Transfer.** For synthetic-to-Real image transfer, we use the VisDA 2017 dataset for evaluation. In total, VisDA 2017 contains over 280k images across 12 categories in the combined training, validation, and testing domains. The training domain consists of 152k synthetic images which are generated by rendering 3D models of the same object categories from different angles and under different lighting conditions. The validation domain includes 55k images by cropping object in real images from COCO. And the testing domain consists of 72k images cropped from video frmaes in YT-BB. As the labels for the testing data are unavailable, we use the training data as our source domain and the validation data as our target domain. Following Pan et al. (2019), we adopt 50-layer ResNet pre-trained on ImageNet as our basic CNN architecture.

**Office-Home Transfer.** (Venkateswara et al., 2017) consists of 15,500 images in 65 object classes in office and home settings, forming four extremely dissimilar domains: Artistic images (Ar), Clip Art (Cl), Product images (Pr), and Real-World images (Rw). By permuting the 4 domains, we can generate 12 different domain adaptation tasks. Our method perform the best in all the transfer tasks compared to state-of-the-art UDA methods. The experiment results and training details for this experiment are in the Appendix A.3 due to space constraint.

**Implementation Details.** Following Wan et al. (2018), we assume the covariance matrices $\{\mathbf{\Sigma}_c^s\}_{c=1}^C$ to be diagonal and the prior probability to be $p(c) = 1/C$ when pre-training the network on the labeled source data. The three trade-off parameters $\varphi$, $\alpha$ and $\beta$ in Eqn. 7 are simply set to be 0.1, 1, 0.1. We strictly follow Pan et al. (2019) and set the embedding dimension $d$ as 10/512 for Digits/synthetic-to-real image transfer. We implement DMPN with Pytorch. We use ADAM with 0.0005 weight decay and 0.9/0.999 momentum for training and set the mini-batch size to be 128/120 in Digits/synthetic-to-real image transfer. We train the network for 350 epochs for the Digits Image transfer tasks. The learning rate is initially set to be 1e-5 for the covariance matrices and 1e-3 for the other parameters[4] and is decayed by 0.1 at epoch 150 and 250. For the synthetic-to-real image transfer, we fix the learning rate to be 1e-6 and train the network for 100 epochs[4]. Finally, for the Digits Image transfer tasks, we apply weighted PDM loss to remedy the labeling error, where each sample is weighted by the maximum predicted class probability. For the synthetic-to-real image transfer task, we apply filtering to remedy the labeling error, where only target examples with maximum predicted probability over 0.8 is used for training. Following the standard, for Digits Image transfer tasks, we adopt the classification accuracy on target domain as evaluation metric and for synthetic-to-real image transfer, we use the average per class accuracy for evaluation metric. **We will publish our code upon acceptance.**

**Compared Methods.** To demonstrate the benefits of our proposed method, we compare it with the following approaches: (1) **Source-only** directly exploits the classification model trained on source domain to classify target samples. (2) **RevGrad** (Ganin & Lempitsky, 2014) trains domain invariant features via adding a domain classifier in the deep feature learning pipeline via gradient reversal. (3) **DC** (Tzeng et al., 2014) minimizes MMD along with deep feature learning for domain invariant features. (4) **DAN** (Long et al., 2015) applies multiple variants of MMD to align feature representations from multiple layers. (5) **RTN** (Long et al., 2016) extends DAN by adapting classifiers through a residual transfer module. (6) **ADDA** (Tzeng et al., 2017) separates the source feature learning and target feature learning using different networks and use a domain discriminator to learn domain invariant features. (7) **JAN** (Long et al., 2017) aligns the joint distribution of the network

---

[3]For the transfer S → M, we insert batch normalization layers after each layer in the original architecture (2 conv-layer LeNet), as the model fails to converge without them. For fair comparison, we also add batch normalization layers for other compared method.

[4]Learning rates are chosen based on the validation results from pre-training on labeled source data.

activation of multiple layers across domains. (8) **MCD** (Saito et al., 2018) employs task-specific decision boundaries to align the distributions of source and target domains. (9) **CDAN+E** (Long et al., 2018) adds a conditional adversarial classifier on the deep feature learning pipeline to learn domain invariant features. (10) **S-En+Mini-aug** (French et al., 2018) modifies the mean teacher variant of temporal ensembling for UDA. (11) **TPN** (Pan et al., 2019) is the first work to apply PN for UDA. **TPN**$_{gen}$ is the variant trained only with general-purpose domain discrepancy loss. (12) **DMPN** is our proposed method. **DMPN**$_{GCMM}$ and **DMPN**$_{PDM}$ are trained only with GCMM loss and PDM loss respectively. (13) **Train-on-target** is an oracle that trained on labeled target samples.

Table 1: Classification accuracy (%) of different methods for (a) Digits Image transfer across MNIST (M), USPS (U) and SVHN (S), and (b) Synthetic-to-real image transfer on VisDA 2017 dataset. We draw the results directly from the original paper if experimental settings are the same. For Digits Image transfer, results with '*' are trained without batch normalization layers (See Section 4.2 for detail). For synthetic-to-real image transfer, we only report the mean accuracy due to space constraint. The detailed per class accuracy results are available in the Appendix A.1

(a) Digits Image transfer

| Method | M $\to$ U | U $\to$ M | S $\to$ M |
|---|---|---|---|
| Source-only | 75.2 | 57.1 | 64.4 |
| RevGrad | 77.1 | 73.0 | 73.9* |
| DC | 77.1 | 73.0 | 67 |
| DAN | 80.3 | 77.8 | 73.7 |
| RTN | 82.0 | 81.2 | 75.3* |
| ADDA | 89.4 | 90.1 | 83.6 |
| JAN | 84.4 | 83.4 | 78.4* |
| MCD | 90.0 | 88.5 | 81.8 |
| TPN$_{gen}$ | 91.3 | 93.5 | 90.2* |
| TPN | 92.1 | 94.1 | 93.0* |
| DMPN$_{GCMM}$ | 94.3 | 94.6 | 96.5 |
| DMPN$_{PDM}$ | 91.8 | 89.1 | 96.6 |
| DMPN | **94.7** | **94.8** | **96.8** |
| Train-on-target | 97.3 | 96.7 | 99.3 |

(b) Synthetic-to-real image transfer

| Method | Mean Accuracy |
|---|---|
| Source-only | 55.3 |
| RevGrad | 58.6 |
| DC | 55.5 |
| DAN | 59.8 |
| RTN | 63.8 |
| JAN | 66.5 |
| CDAN+E | 70.0 |
| MCD | 71.9 |
| S-En+Mini-aug | 74.2 |
| TPN$_{gen}$ | 73.6 |
| TPN | 80.4 |
| DMPN$_{GCMM}$ | 80.2 |
| DMPN$_{PDM}$ | 81.3 |
| DMPN | **81.4** |
| Train-on-target | 95.8 |

## 4.2 EXPERIMENTAL ANALYSIS

**Performance Comparison.** Table 1 shows the results of all methods for the two tasks. Overall, our proposed method achieves superior results than all the existing methods. For the Digits Image transfer tasks, DMPN has improved the accuracy for M $\to$ U, U $\to$ M and S $\to$ M by 2.6%, 0.7% and 3.8% respectively compared to the second best. We have made great advancement considering the second best accuracy results are already quite high. For the task S $\to$ M, due to convergence reasons, we have added batch normalization layers to the original CNN architectures. For fair comparison, we have re-run some experiments on other methods by adding batch normalization layers to them. For methods whose public code are not available, we simply report the accuracy results with the original CNN architecture. For ADDA, adding batch normalization layer has improved its accuracy result from 76.0% to 83.6%, which has an increase of 7.6% of accuracy. However, we doubt adding batch normalization layers will have the same effect on TPN, as TPN already has a quite high accuracy. Nonetheless, we think our accuracy result of 96.8% on this task will be difficult for the other methods to surpass even with batch normalization layers. For the Synthetic-to-real image transfer task, we only compare with methods without extensive data augmentations and our method has increased the state-of-the-art single model mean accuracy by 1.0%. TPN$_{gen}$ reduces the domain discrepancy via minimizing the pairwise Reproducing Kernel Hilbert Space (RKHS) distances among the corresponding prototypes of the source-only, target-only and source-target data. In DMPN$_{GCMM}$ we minimize the $L^2$ distance between the corresponding Gaussian component means of the source and target data. The $L^2$ distance can be viewed as the distance in a Linear RKHS space. The calculation of our proposed GCMM loss is much simpler than the general purpose loss, yet with explicitly modeling of the feature distributions, DMPN$_{GCMM}$ has a gain of accuracy of 3.0%, 1.1%, 6.3% and 6.6%

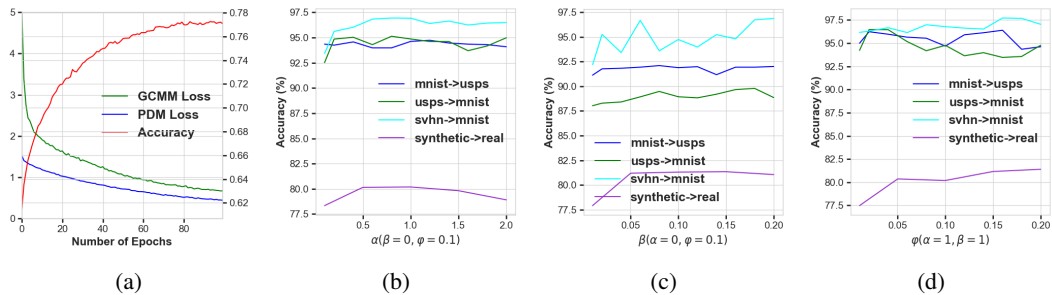

Figure 2: (a): GCMM loss, PDM loss and Accuracy with the increase of training epochs on VisDA. (b)-(d): Sensitivity analysis on hyper-parameters $\alpha$, $\beta$ and $\varphi$ respectively.

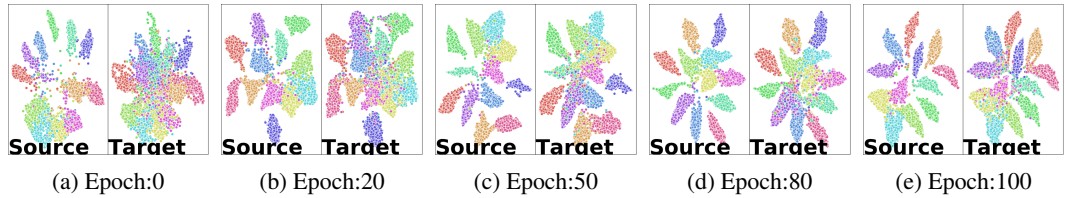

| (a) Epoch:0 | (b) Epoch:20 | (c) Epoch:50 | (d) Epoch:80 | (e) Epoch:100 |

Figure 3: (a)-(e): The t-SNE visualizations of features generated by DMPN with the increase of training epochs on VisDA 2017.

on the $M \rightarrow U$, $U \rightarrow M$, $S \rightarrow M$ and synthetic-to-real image transfer tasks respectively compared with TPN$_{gen}$, showing that our method is more effective.

**Ablation Analysis.** In Table 1, combining GCMM loss and PDM loss helps to increase the accuracy results, showing that the two domain discrepancy losses are compatible to each other. DMPN$_{GCMM}$ performs better than or similar to almost all other domain adaptation methods and DMPN$_{PDM}$ performs better than most of them.

**Convergence Analysis.** Figure 2 (a) shows the training progress of DMPN. The GCMM loss and PDM loss keep decreasing with more training epochs. The prediction accuracy on the unlabeled target data keeps increasing. And the noise ratio $\rho$ decreases along the training process, from the initial value of 38.6% decreases to 22.9%, which supports our theoretical analysis in Section 3.5. Figure 3 shows the t-SNE visualizations of the source and target embedded features during training, which shows that target classes are becoming increasingly well discriminated by the source classifier.

**Sensitivity Analysis.** Figure 2 (b)-(d) shows the sensitivity analysis on the hyper-parameters $\alpha$, $\beta$ and $\varphi$ with the other hyper-parameters fixed. Overall, the experiment results show that we can get similar accuracy results or even better when changing the hyper-parameters in a certain range, demonstrating that our method is robust against hyper-parameter changes. The sensitivity analysis on the confidence threshold is in the Appendix A.2, which shows our method is robust against threshold value.

## 5 CONCLUSIONS

In this paper, we propose Distribution Matching Prototypical Network (DMPN) for Unsupervised Domain Adaptation (UDA) where we explicitly model and match the deep feature distribution of the source and target data as Gaussian mixture distributions. Our work fills the gap in UDA where state-of-the-art methods assume the deep feature distributions of the source and target data are unknown when minimizing the discrepancy between them. We propose two new domain discrepancy losses based on the Gaussian mixture distributions of the deep features called Gaussian Component Mean Matching (GCMM) and Pseudo Distribution Matching (PDM). Extensive experiments verify the effectiveness of our proposed method and domain discrepancy losses. Finally, a post-hoc hyper-parameter sensitivity analysis shows that our approach is robust w.r.t hyper-parameter changes.

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

# A APPENDIX

## A.1 DETAILED RESULTS ON SYNTHETIC-TO-REAL IMAGE TRANSFER

Table 2: Classification accuracy (%) of different methods for Synthetic-to-real image transfer on VisDA 2017 dataset

| Method | plane | bcycl | bus | car | horse | knife | mcycl | person | plant | sktbrd | train | truck | mean |
|---|---|---|---|---|---|---|---|---|---|---|---|---|---|
| Source-only | 70.6 | 51.8 | 55.8 | 68.9 | 77.9 | 7.6 | 93.3 | 34.5 | 81.1 | 27.9 | 88.6 | 5.6 | 55.3 |
| RevGrad | 75.9 | 70.5 | 65.3 | 17.3 | 72.8 | 38.6 | 58.0 | 77.2 | 72.5 | 40.4 | 70.4 | 44.7 | 58.6 |
| DC | 63.6 | 38.4 | 71.2 | 61.4 | 71.4 | 10.9 | 86.6 | 43.5 | 70.2 | 47.7 | 79.8 | 21.6 | 55.5 |
| DAN | 61.7 | 54.8 | 77.7 | 32.2 | 75.0 | 80.8 | 78.3 | 46.9 | 66.9 | 34.5 | 79.6 | 29.1 | 59.8 |
| RTN | 79.5 | 59.6 | 78.0 | 47.4 | 82.7 | 82.0 | 84.7 | 54.7 | 81.6 | 34.5 | 74.2 | 6.6 | 63.8 |
| JAN | 92.1 | 66.4 | 81.4 | 39.6 | 72.5 | 70.5 | 81.5 | 70.5 | 79.7 | 44.6 | 74.2 | 24.6 | 66.5 |
| MCD | 87.0 | 60.9 | **83.7** | 64.0 | 88.9 | 79.6 | 84.7 | 76.9 | 88.6 | 40.3 | 83.0 | 25.8 | 71.9 |
| TPN$_{gen}$ | 94.5 | **86.8** | 76.8 | 49.7 | **92.1** | 12.5 | 84.7 | 75.2 | 92.1 | 86.8 | 84.1 | 47.4 | 73.6 |
| TPN$_{task}$ | 89.2 | 62.8 | 71.7 | **83.5** | 90.6 | 24.6 | 88.8 | **91.1** | 89.8 | 74.7 | 69.1 | 36.1 | 72.7 |
| TPN | 93.7 | 85.1 | 69.2 | 81.6 | 93.5 | 61.9 | **89.3** | 81.4 | **93.5** | 81.6 | 84.5 | 49.9 | 80.4 |
| DMPN$_{GCMM}$ | 94.1 | 83.0 | 73.7 | 54.7 | 90.7 | **89.2** | 80.5 | 79.1 | 91.0 | 89.9 | 82.8 | 53.2 | 80.2 |
| DMPN$_{PDM}$ | **94.8** | 85.4 | 80.0 | 55.5 | 90.9 | 84.6 | 78.7 | 78.6 | 90.7 | **92.4** | **88.8** | 55.5 | 81.3 |
| DMPN | 94.6 | 84.9 | 75.7 | 57.5 | 91.2 | 88.1 | 80.6 | 78.6 | 91.3 | 91.6 | 86.7 | **55.6** | **81.4** |
| S-En+Mini-aug | 92.9 | 84.9 | 71.6 | 41.2 | 88.8 | 92.4 | 67.5 | 63.5 | 84.5 | 71.8 | 83.2 | 48.1 | 74.2 |
| S-En+Test-aug | 96.3 | 87.9 | 84.7 | 55.7 | 95.9 | 95.2 | 88.6 | 77.4 | 93.3 | 92.8 | 87.5 | 38.2 | 82.8 |
| Train-on-target | 99.5 | 91.9 | 97.3 | 96.8 | 98.3 | 98.5 | 94.1 | 96.2 | 99.0 | 98.2 | 97.9 | 82.3 | 95.8 |

## A.2 SENSITIVITY ANALYSIS ON CONFIDENCE THRESHOLD

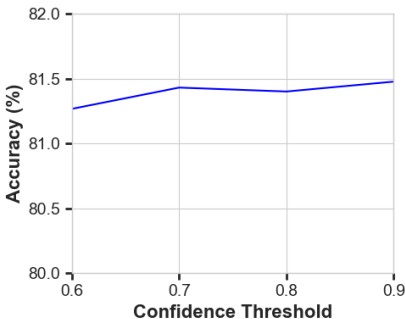

Figure 4: Sensitivity analysis on confidence threshold.

Fig. 4 shows the sensitivity analysis of our method on different values of confidence threshold on VisDA 2017 dataset. The experiment results show that we can get similar accuracy results or even better when changing the confidence threshold in a certain range, demonstrating that our method is robust against hyper-parameter changes.

## A.3 OFFICE-HOME TRANSFER

Table 3 presents experiment results of state-of-the-art UDA methods and our method on Office-Home dataset. Our method gives the best accuracy results in all transfer tasks, showing the effectiveness of our method. In this experiment, we train the network for 100 epochs. The learning rate is initially set to be 1e-5 for all the parameters and is decayed by 0.1 at epoch 60 and 80. Following Long et al. (2018), $\alpha$ and $\beta$ vary in each training epoch and are calculated by $\alpha = \frac{1-e^{-\gamma p}}{1+e^{-\gamma p}}$ and $\beta = 0.1 * \frac{1-e^{-\gamma p}}{1+e^{-\gamma p}}$ respectively, where $\gamma$ is set to be the default value 10, $p$ is the training process changing from 0 to 1.

Table 3: Classification accuracy (%) of different methods on Office-Home dataset

| Method | Ar→Cl | Ar→Pr | Ar→Rw | Cl→Ar | Cl→Pr | Cl→Rw | Pr→Ar | Pr→Cl | Pr→Rw | Rw→Ar | Rw→Cl | Rw→Pr | Avg |
|---|---|---|---|---|---|---|---|---|---|---|---|---|---|
| Source-only | 34.9 | 50.0 | 58.0 | 37.4 | 41.9 | 46.2 | 38.5 | 31.2 | 60.4 | 53.9 | 41.2 | 59.9 | 46.1 |
| RevGrad | 45.6 | 59.3 | 70.1 | 47.0 | 58.5 | 60.9 | 46.1 | 43.7 | 68.5 | 63.2 | 51.8 | 76.8 | 57.6 |
| DAN | 43.6 | 57.0 | 67.9 | 45.8 | 56.5 | 60.4 | 44.0 | 43.6 | 67.7 | 63.1 | 51.5 | 74.3 | 56.3 |
| JAN | 45.9 | 61.2 | 68.9 | 50.4 | 59.7 | 61.0 | 45.8 | 43.4 | 70.3 | 63.9 | 52.4 | 76.8 | 58.3 |
| CDAN | 49.0 | 69.3 | 74.5 | 54.4 | 66.0 | 68.4 | 55.6 | 48.3 | 75.9 | 68.4 | 55.4 | 80.5 | 63.8 |
| CDAN+E | 50.7 | 70.6 | 76.0 | 57.6 | 70.0 | 70.0 | 57.4 | 50.9 | 77.3 | 70.9 | 56.7 | 81.6 | 65.8 |
| TPN | **53.4** | **71.4** | **76.1** | **59.0** | **70.8** | **70.5** | **61.8** | **54.3** | **80.7** | **71.5** | **58.4** | **84.4** | **67.7** |

