# OpenReview forum: "Distribution Matching Prototypical Network for Unsupervised Domain Adaptation"
_ICLR.cc/2020/Conference — Reject_

### Official Review · AnonReviewer3 · 2019-10-23
**Official Blind Review #3**

**Rating:** 3

**Review:**

<Paper summary>
The authors proposed Distribution Matching Prototypical Network (DMPN) for unsupervised domain adaptation. DMPN extracts features from the input data and models them as Gaussian mixture distributions. By explicitly modeling the distributions that the features follow, the discrepancy between the distribution of source data and that of target data can be easily evaluated. DMPN is trained by jointly minimizing two kinds of loss, which are classification loss on the source data and domain discrepancy loss that is calculated via the explicit models. Experimental results on two popular benchmark datasets validate the advantage of DMPN over other state-of-the-art methods.

<Review summary>
The proposed method seems simple but empirically performs well. The paper is well written and easy to follow, so we can maybe easily implement it. However, I have several concerns mainly about the details and theories of the proposed method, which makes my score a bit lower than the border line. Given clarifications in an author response, I would be willing to increase the score.

<Details>
* Strength
 + The motivation of using ProtoNet for domain adaptation seems reasonable.
 + The proposed method performs well in the experiments.
 + The paper, especially the experiment section, is well written and easy to follow.


* Weakness and concerns
 - Several points on the proposed loss (GCMM and PDM) are not sufficiently discussed.
  -- Why do we need two kinds of loss? These losses seem to play almost same role. Since PDM loss corresponds to target-side log likelihood regularization term (Eq. (3)), I wonder if we really need GCMM loss.
  -- Since the authors explicitly model the feature distributions by Gaussian mixtures (GMs), it might be possible to calculate a standard divergence between source and target data distributions by using the parameters of GMs. Compared with such a straightforward approach, the proposed method seems to be ad-hoc and is not theoretically validated. What term of divergence (or distance) does it minimize?
  -- When a certain class does not appear in pseudo-labeled target data, how can we calculate GCMM loss? (specifically, \mu^{et}_c)
  -- Are Eq. (3) and Eq. (6) correct? These are defined as total loss, not average, over each domain. It means that the scale of the coefficients for these terms changes according to the number of training data, but the sensitivity analysis in Fig. 2 does not show such effect.
  -- Since the proposed losses heavily depend on the pseudo labels on the target data, it should be important to carefully set a proper threshold for the confidence. Is the proposed method sensitive against the change of this threshold? If so, how can we tune it?
  -- How can we know p(c) in advance?

 - The theory shown in 3.5 is not sufficiently validated.
  -- The authors state ````we minimize the first term through minimizing the domain discrepancy losses," but it is not sufficiently supported, because the relationship between the proposed losses and H-delta-H divergence is not clear.

 - I am concerned about whether the proposed method works well with harder datasets such as Office-Home dataset, because each class data are modeled by a simple Gaussian distribution in the proposed method.


* Minor concerns that do not have an impact on the score
 - Using both f^s_i and F(x^s_i; \theta) is confusing.
 - Typo in Eq. (7): PMD -> PDM


**Experience Assessment:**

I have published in this field for several years.

**Review Assessment: Checking Correctness Of Derivations And Theory:**

I assessed the sensibility of the derivations and theory.

**Review Assessment: Checking Correctness Of Experiments:**

I assessed the sensibility of the experiments.

**Review Assessment: Thoroughness In Paper Reading:**

I read the paper thoroughly.

---

> ### Author Response · Authors · 2019-11-07
> **Response to Reviewer #3**
>
> Thanks for reviewing our paper and your appreciation of our idea. Here we answer your concerns and clarify some of the weak points you mentioned:
>
> 1). These two losses actually serve with different purposes when we design them. The GCMM loss brings the two distribution closer via minimizing the corresponding Gaussian Component means of the source and target data. And the PDM loss shapes the target feature distribution similar as the source feature distribution via minimizing the likelihood of generating the target feature from the source feature distribution. In this sense, they complement each other, to match the target feature distribution to be exactly like the source feature distribution. Furthermore, these two losses also reduce distribution discrepancy at different levels, GCMM reduces distribution discrepancy at the class-level and PDM reduces distribution discrepancy at the sample-level, thus in this sense, they also complete each other for domain adaptation.
>
> 2). We want to clarify here. Our method does not learn the distribution parameters for the target data. We learn the distribution parameters of the source data. We use the empirically calculated distribution parameter estimator of the source and target data to minimize the distribution discrepancy loss function. Thus, we cannot "calculate a standard divergence between source and target data distributions by using the parameters of GMs." For the GCMM loss, our method minimizes the euclidean distance between the corresponding Gaussian Component means of the source and target data for each class. PDM loss minimizes the likelihood of generating the target feature with the source feature distribution.
>
> 3). We will ignore data from that class in the batch in that training iteration. As training data are sampled randomly in each iteration, and in the end, all data updates the model.
>
> 4). Yes, you are correct. We forgot to average the term when writing the paper. We have corrected it in the revised paper. Thanks for pointing this out.
>
> 5). We have added a sensitivity experiment on the confidence threshold. The results are in the appendix of the revised paper. Here is the summary:
>
> confidence-threshold: 0.6, 0.7, 0.8, 0.9
> Mean-accuracy: 81.3, 81.4, 81.4, 81.5
>
> The results show that our method is also robust against confidence threshold.
> For our proposed probability based weighting mechanism, as there is no hyper-parameter in there, so there is no need to provide sensitivity analysis on it.
>
> 6). We know p(c) in the source domain, as it has labels. We do not know p(c) in the target domain, but we can estimate it. In this paper, we assume p(c) is uniform, as we focus on co-variate shift in this paper. Our work can be easily augmented to work for label shift too, once we estimate the target label distribution. However, we leave this as future work.
>
> 7). The H-delta-H divergence is small when the two distribution discrepancy is small. As GCMM loss brings the two distribution closer, and PDM loss shapes the two distribution to be alike, the source and target feature distribution discrepancy will be smaller. Thus H-delta-H becomes smaller as we minimize GCMM loss and PDM loss. We have updated the paper on this part to make it clearer. Thanks for indicating this.
>
> 8). We have added an experiment on the Office-Home dataset in the appendix of our paper. Our paper performs the best in all the transfer tasks in Office-Home compared to state-of-the-art UDA methods, showing that it also works for this more challenging dataset.
>
> 9). Thanks for pointing out some of our typos, we have made the changes in our revised paper.

---

> > ### Comment · AnonReviewer3 · 2019-11-08
> > **Thank you for your response**
> >
> > Thank you for your response.
> >
> > 2) 4) 5) 8) 9) make sense to me. Especially, thank the authors for the additional experimental result on the Office-Home dataset.
> >
> > 1) I got the intention of the authors, but I am not yet convinced with how the losses work. I want to confirm the following points one by one.
> >  - Are the source and target distributions perfectly matched if and only if the both losses are ideally minimized?
> >  - Can it be achieved only when we use both losses?
> >   -- If yes, what is the difference between the two losses at the optimal point?
> >   -- If no, why do we need both? If it affects the optimization dynamics (not the goal of the optimization), could you explain it intuitively?
> >
> > 3) I want to confirm my understanding on how to calculate GCMM loss. Since mu_s and mu_t are the averaged features over the all data of the respective domain, I was thinking that they are initialized after the pre-training and are updated by using each mini-batch like stats in the batch-normalization layer. Is it correct? If so, when a certain class does not appear in pseudo-labeled target data after the pre-training, and we cannot calculate GCMM loss due to lack of the initial mu of that class.
> >
> > 6) Assuming covariate shift does not support why we can assume the uniform class prior at the target domain. But, anyway, I understand that this problem is remained in the future works.
> >
> > 7) I am asking what is "the two distribution discrepancy" in the proposed method. For example, adversarial training in the original GAN minimizes JS divergence. Since JS divergence can bound L1 distance that can bound the H-delta-H divergence [R1], using the adversarial training for domain adaptation is meaningful from the perspective of Ben David's paper. The statement in 3.5 justifies that we can use noisy target labels, not the proposed losses, because the authors do not clarify the relationship between the proposed losses and H-delta-H divergence. And, the justification of using the noisy target labels seems to be already shown in [R2].
> >
> > [R1] Domain Adaptation: Learning Bounds and Algorithms, COLT 2009
> > [R2] Asymmetric Tri-training for Unsupervised Domain Adaptation, ICML 2017

---

> > > ### Author Response · Authors · 2019-11-08
> > > **Response to Reviewer #3 Part 2**
> > >
> > > 7). a). First, let us set the ground, we can have different measurements of distribution discrepancy. H-delta-H divergence is one measurement for distribution discrepancy and the larger the distribution discrepancy the larger the H-delta-H divergence.
> > >
> > > In our paper, we model the feature distribution as Gaussian mixture.
> > > GCMM is a measurement of the distribution discrepancy between two Gaussian mixture distributions. It measures the distance between the coresponding Gaussian component means of two Gaussian mixture distributions. If two Gaussian mixture distributions have quite different Gaussian component means, then GCMM will be large.
> > >
> > > PDM is another measurement of the distribution discrepancy between two distributions. It measures the negative log likelihood of data points from P1 on P2. The larger the distribution discrepancy, the larger the PDM loss.
> > >
> > > Now, we have three measurements for distribution discrepancy. Intuitively, these measurements are all positively related to distribution discrepancy. Thus, reducing two of them will also reduce the third one. Thus we claim "The H-delta-H divergence is small when the two distribution discrepancy is small.".
> > >
> > > To make it more formal or make it more clearer. Let us introduce the forth measurement, the L1-distance bewteen two distributions as the default one.
> > >
> > > Now, clearly when L1-distance between two distributions is small, H-delta-H divergence is small;
> > > minimizing GCMM loss reduces the L1-distance between two Gaussian mixtures, as the two distributtions are moved closer;
> > > minimizing PDM loss also reduces the L1-distance between two Gaussian mixtures, as data points from one distribution are moved closer to another distribution;
> > > Thus, we can conclude, "The H-delta-H divergence is small when the two distribution discrepancy is small."
> > >
> > > Sorry if there are too many words to read, we just want to make the question clearer.
> > >
> > > The above is our full justification why minimizing GCMM and PDM reduces H-delta-H divergence.
> > >
> > > Our clarification of the relationship between the two distribution discrepancy losses and the H-delta-H divergence is updated in the paper as "In DMPN, we minimize the first term through minimizing the domain discrepancy losses, as H-delta-H is small when the source features and target features have similar distribution and minimizing the domain discrepancy losses makes the source and target feature to distribute similarly."
> > >
> > > Thanks.
> > >
> > > b). Our justification of using noisy target labels is different from the justification in [2]. Our justification is derived based on the theory in Supervised Domain Adaptation (Lemma 4 in [3]), while Saito et al.'s is based on Unsupervised Domain Adaptation (Theorem 2 in [3]). As we are trying to justify using noisy labeled target data, so deriving the justification from Supervised Domain Adaptation is more proper.
> > >
> > >
> > >
> > >
> > > [1] Lipton, Z. C., Wang, Y. X., & Smola, A. (2018). Detecting and correcting for label shift with black box predictors. arXiv preprint arXiv:1802.03916.
> > > [2] Saito, K., Ushiku, Y., & Harada, T. (2017, August). Asymmetric tri-training for unsupervised domain adaptation. In Proceedings of the 34th International Conference on Machine Learning-Volume 70 (pp. 2988-2997). JMLR. org.
> > > [3] Ben-David, S., Blitzer, J., Crammer, K., Kulesza, A., Pereira, F., & Vaughan, J. W. (2010). A theory of learning from different domains. Machine learning, 79(1-2), 151-175.

---

> > > > ### Comment · AnonReviewer3 · 2019-11-11
> > > > **Thank you for your response**
> > > >
> > > > 7) a) "when L1-distance between two distributions is small, H-delta-H divergence is small" is proved in [R1], but "minimizing GCMM loss reduces the L1-distance" and "minimizing PDM loss also reduces the L1-distance" are not mathematically proved in this paper. These proofs are necessary to obtain the bound of the target error in the proposed method.
> > > >
> > > > 7) b) Thank you. I understand the difference.

---

> > > ### Author Response · Authors · 2019-11-08
> > > **Response to Reviewer #3 Part 1**
> > >
> > > Thank you for your response. Here we answer the questions you have mentioned:
> > >
> > > 1). a). Ideally minimizing the two discrepancy losses is a sufficient condition for exact feature distribution matching between the source and target data but is not a necessary condition. The first thing we need to keep in mind is that the source feature distribution is not fixed during training. So if we are able to ideally minimize the two losses, then it is natural to assume that the classification loss on the labeled source data can also be ideally minimized. In that case, the source feature Gaussian mixture distribution collapse to N feature vectors, where each vector represents the Gaussian component mean for one class and all source data in the same class will have the exact same feature representation as the Gaussian component mean for that class. If the two discrepancy losses are ideally minimized, then the target feature distribution also collapses to the N feature vectors, which is an exact feature distribution matching between the source and target data. Thus ideally minimizing the two discrepancy losses ensures exact feature distribution matching between the source and target data. Combining the classification loss and the two discrepancy loss fucntions, this is exactly the objective function our method is trying to minimize. Thus, our method approaches exact feature distribution matching between the source and target data as we minimize its objective.
> > >
> > > Suppose we have exact feature distribution matching between the source and target data. Then the GCMM loss is ideally minimized, which is 0. If we assume the source feature distribution does not collapse to the N component means, then PDM loss can be further minimized until the target feature distribution collapse into the N Gaussian component mean vectors. Thus, exact feature distribution matching does not prove the two discrepancy loss functions are ideally minimized. However, there is no need to worry here, as keep minimizing the PDM loss maintains the decision boundary. Our method, which keeps minimizing the PDM loss, will not result in accuracy drop for the target data.
> > >
> > > b). No, theoretically by only minimizing the PDM loss can achieve exact feature distribution matching. As shown in our experiment results, DMPN_PDM already performs quite well.
> > >
> > > The reason why we need both is that they work at different levels. GCMM brings the entire target Gaussian component closer to the source Gaussian component and PDM generates target features closer to the source feature distribution. GCMM works at the class level and PDM works at the sample level. In some sense, they complement each other. In the extreme case when each class has only one data point, GCMM loss reduces to PDM loss, thus they are not conflicting each other but only working at different levels to achieve the same goal.
> > >
> > > The optimal point for GCMM loss, is when the Gaussian component means of the target features coincide with the source feature Gaussian component means. Thus minimizing the GCMM loss, pulls the target Gaussian component means towards the source Gaussian component means, which helps to decrease the PDM loss. Similarly, minimizing the PDM loss ensures the embedding function to generate the target features closer to the source Gaussian component means, which in turn minimize the GCMM loss. Therefore, minimizing the two discrepancy losses boost each other and will not result in bad behavior for optimization.
> > >
> > > 3). No, we do not keep global statistical estimators for mu_s and mu_t like batch-normalization layer. The distribution parameters for source data is learned automatically by back-propagation, which includes mu_s. We do not need global mu_t for the target data. We estimate mu_t in each mini-batch.
> > >
> > > 6). If the co-variate shift assumption does not hold, we may assume uniform class prior when the label shift is mild. We cannot assume uniform class prior when the label shift is severe, as severe label shift will degrade the performance of the transfered model in the target domain [1] by a lot.

---

> > > > ### Comment · AnonReviewer3 · 2019-11-11
> > > > **Thank you for your response**
> > > >
> > > > 1) a) Thank you for your thorough explanation. I understand that minimizing the two losses is a sufficient condition for distributional matching.
> > > >
> > > > 1) b) So, using PDM loss is enough in the ideal case, and GCMM loss works like a kind of regularization for PDM loss, while it does not change the goal. "Working at different levels" sounds good, but it is quite better if the authors can show a good example in which minimizing PDM loss is not enough and GCMM loss helps. (reverse case is shown in the authors' response of 1-a)
> > > >
> > > > 3) Sorry for typos. They should be mu^{es} and mu^{et}. Are they estimated in each mini-batch? If so, does the computed GCMM loss result in the average of the distance between the means? It seems to depend on the class priors, but it does not in Eq. (5).

---

### Official Review · AnonReviewer1 · 2019-10-25
**Official Blind Review #1**

**Rating:** 3

**Review:**

The paper develops a new method for adapting models trained on labeled data from some source domain to unlabeled data in a target domain. The authors accomplish this by adapting a technique from [1] and [2] enforcing that the deep features learned during training approximately follow a Gaussian mixture distribution. With the learned features in this form, the authors ensure domain adaptation by minimizing the discrepancy between the distributions arising from the source and target datasets.

Strengths:
 + The paper's experiments show an improvement in the model's performance relative to past work, utilizing a large number of comparison models.
 + The use of explicit distributional information within the learned representations seems like a good fit for the task at hand, and the authors' experiments back this up.

Weaknesses:
 - The proposed method for unsupervised domain adaptation is very similar to the prototypical networks approach in [3], with the primary difference being a loss term incentivizing a Gaussian mixture distribution over features.
 - While the authors achieve improved performance over [3], the gains in classification accuracy on the target dataset aren't especially huge (~1-3%).
 - The paper is a bit hard to follow, and would be improved by giving a more explicit comparison of the methods used here to past work, especially [1] and [3].


[1] Weitao Wan, Yuanyi Zhong, Tianpeng Li, and Jiansheng Chen. Rethinking feature distribution for loss functions in image classification. 2018 IEEE/CVF Conference on Computer Vision and Pattern Recognition, pp. 9117–9126, 2018.

[2] Hong-Ming Yang, Xu-Yao Zhang, Fangying Yin, and Chenglin Liu. Robust classification with convolutional prototype learning. 2018 IEEE/CVF Conference on Computer Vision and Pattern Recognition, pp. 3474–3482, 2018.

[3] Yingwei Pan, Ting Yao, Yehao Li, Yu Wang, Chong-Wah Ngo, and Tao Mei. Transferrable prototypical networks for unsupervised domain adaptation. In Proceedings of the IEEE Conference on Computer Vision and Pattern Recognition, pp. 2239–2247, 2019.

**Experience Assessment:**

I do not know much about this area.

**Review Assessment: Checking Correctness Of Derivations And Theory:**

I assessed the sensibility of the derivations and theory.

**Review Assessment: Checking Correctness Of Experiments:**

I assessed the sensibility of the experiments.

**Review Assessment: Thoroughness In Paper Reading:**

I read the paper at least twice and used my best judgement in assessing the paper.

---

> ### Author Response · Authors · 2019-11-07
> **Response to Reviewer #1 Part 2**
>
> 3). For your advice of adding more explicit comparison between our work and [1,7], we have added some comparisons in the paper, hopefully it will make the paper clearer.
>
> As suggested by one of the reviewers, we have provided further sensitivity analysis of our method on the confidence threshold. Our default confidence threshold value is set to be 0.8. And we have experimented it with some other confidence values, 0.6, 0.7 and 0.9. The experiment results are on the revised paper. Please check the results in Figure 4 in the Appendix. The results show our method is also robust against confidence threshold value.
>
> As also suggested by one of the reviewers, we have provided further experiment on the more challenging Office-Home dataset. Our method performs the best in all transfer tasks than state-of-the-art UDA methods. The results are in our revised paper (Table 3 in the Appendix), you are welcomed to check that out.
>
> Finally, after our clarifications, we hope you have a better understanding of our work and give a more fair grade to our work. Thanks.
>
> [1] Yingwei Pan, Ting Yao, Yehao Li, Yu Wang, Chong-Wah Ngo, and Tao Mei.  Transferrable proto-typical networks for unsupervised domain adaptation. InProceedings of the IEEE Conference on Computer Vision and Pattern Recognition, pp. 2239–2247, 2019.
> [2] Jake Snell, Kevin Swersky, and Richard Zemel. Prototypical networks for few-shot learning. InAdvances in Neural Information Processing Systems, pp. 4077–4087, 2017.
> [3] Mingsheng Long, Yue Cao, Jianmin Wang, and Michael I Jordan.  Learning transferable featureswith deep adaptation networks.arXiv preprint arXiv:1502.02791, 2015.
> [4] Baochen Sun and Kate Saenko.  Deep coral: Correlation alignment for deep domain adaptation.  InEuropean Conference on Computer Vision, pp. 443–450. Springer, 2016.
> [5] Werner  Zellinger,   Thomas  Grubinger,   Edwin  Lughofer,   Thomas  Natschl ̈ager,   and  SusanneSaminger-Platz.   Central moment discrepancy (cmd) for domain-invariant representation learn-ing.arXiv preprint arXiv:1702.08811, 2017.
> [6] Geoffrey  French,  Michal  Mackiewicz,  and  Mark  H.  Fisher.   Self-ensembling  for  visual  domainadaptation. InICLR, 2018.
> [7] Weitao  Wan,  Yuanyi  Zhong,  Tianpeng  Li,  and  Jiansheng  Chen.   Rethinking  feature  distributionfor loss functions in image classification.2018 IEEE/CVF Conference on Computer Vision andPattern Recognition, pp. 9117–9126, 2018.

---

> ### Author Response · Authors · 2019-11-07
> **Response to Reviewer #1 Part 1**
>
> Thanks for reviewing our paper. Here are some points that are not fair to our work based on the weaknesses you have mentioned and we want to argue about them:
>
> 1). There is some connection between our work and the work in [1]. As we have stated in the Related Works section, in [2], "learning PN is equivalent to performing mixture density estimation on the deep features with an exponential density". Thus, modeling the feature distribution as Gaussian Mixture, which is a type of exponential density, is equivalent to learn a Prototypical Network. This statement induces some connection between our work and the work in [1]. However, this equivalence is only true when learning a model in a single domain. Our work is way different from Pan et al.'s work in the setting of domain adaptation.
>
> First, we are based on different ideas. While Pan et al. propose a novel idea to remold PN for domain adaptation, as stated in their paper, our work is based on the idea that almost all existing domain adaptation methods are minimizing the feature distribution discrepancy for effective knowledge transfer from source domain to target domain, but none of them explicitly models the feature distribution though intuitively it facilitates the measuring of distribution discrepancy.
>
> Second, the two works propose different distribution discrepancy loss functions. While Pan et al. propose multi-granular distribution discrepancy loss functions at both class-level and sample-level, our work proposes two new distribution discrepancy loss functions based on probability, one is Gaussian Component Mean Matching (GCMM) and one is Pseudo Distribution Matching (PDM). These two discrepancy loss functions work at different aspects and complement each other, where GCMM brings the two distributions closer, while PDM shapes the two distributions alike. One central component of a domain adaptation method is the distribution discrepancy loss function, as most domain adaptation methods follow a similar framework to minimize the distribution discrepancy loss function together with a classification loss function for knowledge transfer. Thus, you may think our work is very similar to Pan et al.'s. This is because almost all domain adaptation methods follow this similar framework. We do not agree with the claim that "the primary difference between our work and Pan et al.'s is a loss term incentivizing a Gaussian mixture distribution over features." Due to the central role distribution discrepancy loss functions play in a domain adaptation method, devising new distribution discrepancy loss functions is an active research area in domain adaptation [3,4,5]. Please do not ignore the two novel discrepancy loss functions we propose based on our feature distribution modeling.
>
> Third, in terms of training algorithm, our method learns the distribution parameters automatically, while the Pan et al.'s work needs to manually calculate the prototypes for assigning pseudo labels.
>
> Finally, our proposed method fills the important gap in the area of domain adaptation, and to the best of our knowledge, no existing UDA methods have tried to model the feature distribution for domain adaptation.
>
> 2). For the digits image transfer tasks, state-of-the-art results are already quite high, all above 92%, thus a 1~3% of accuracy increase should be considered as significant. There is not much room for a new method to make a huge improvement. If we treat the "Train-on-target" accuracy as the upper bound, the difference of accuracy between the second best results and the upper bound is quite limited, being 5.2%, 2.6%, 6.3% for the transfer M->U, U->M, S->M respectively. For VisDA-2017 dataset, our method improved from the second best by 1%, having a accuracy results of 81.4%.
> And it is 1.4% lower than [6], which won the first place in the VisDA-2017 competition. Thus, although this improvement is not huge, but it should not be considered as marginal.

---

### Official Review · AnonReviewer2 · 2019-10-26
**Official Blind Review #2**

**Rating:** 1

**Review:**

This paper introduces Distribution Matching Prototypical Network (DMPN) for Unsupervised Domain Adaptation (UDA). The proposed method explicitly models the feature distribution as a Gaussian mixture model in both source and target domains. Then the method aligns the target distribution with the source distribution by minimizing losses, which are called Gaussian Component Mean Matching (GCMM) and Pseudo Distribution Matching (PDM).

This paper should be rejected because (1) the novelty of the main idea is marginal, and (2) the performance gain over the baseline methods is also marginal.

Pan et al. already proposed the idea of transferring the knowledge from the source to the target using the prototype of each class. It is required to explain why explicit modeling performs better than implicit modeling of prototypes by theory or practice.

In table 2, the proposed method seems better than TPN, but in the appendix, by comparing then in each category, the proposed method wins six categories, whereas TPN also wins six categories. Therefore, it is hard to say the proposed DMPN is more effective than another method.

Each prototype is modeled using a mean and a covariance matrix. Why the authors don't use the estimated covariance matrix to measure the distance in eq.5?

Because the proposed method uses pseudo-labeling for the target domain, it seems that the weights to determine unreliable examples are crucial. The paper should show the sensitivity of ways to determine the weights. What happens if values of 0.1 and 0.9 are changed in (pi-0.1)/0.9 on page 6?

**Experience Assessment:**

I have published in this field for several years.

**Review Assessment: Checking Correctness Of Derivations And Theory:**

I assessed the sensibility of the derivations and theory.

**Review Assessment: Checking Correctness Of Experiments:**

I assessed the sensibility of the experiments.

**Review Assessment: Thoroughness In Paper Reading:**

I read the paper at least twice and used my best judgement in assessing the paper.

---

> ### Author Response · Authors · 2019-11-07
> **Response to Reviewer #2 Part 2**
>
> For the second reason of rejection, we do not agree as well. For the digits image transfer tasks, state-of-the-art results are already quite high, all above 92%, thus a 1~3% of accuracy increase should be considered as significant. Our method has improved on transfer M->U by 2.6% and on transfer S->M by 3.8% compared to the second best. Taking the results in context, it is not fair to consider these improvements as marginal. For VisDA-2017 dataset, our method improved from the second best by 1%, having a accuracy results of 81.4%. And it is 1.4% lower than [7], which won the first place in the VisDA-2017 competition. Thus, our improvement of 1% in this task should not be considered as marginal either.
>
> For some further questions, you mentioned "Why the authors don't use the estimated covariance matrix to measure the distance in eq.5?" Yes, we have tried that to come up with a correlation distance similar as Deep Coral [5], however it does not perform well, so we do not report it in the paper.
>
> For your question "The paper should show the sensitivity of ways to determine the weights. What happens if values of 0.1 and 0.9 are changed in (pi-0.1)/0.9 on page 6". The value 0.1, and 0.9 are set based on probability, because we have 10 classes for the digits image transfer, so a random prediction would have probability 0.1. If we directly use the probability based weighting, then a random prediction would also contribute to the training, we do not want that to happen, so we weight with (pi-0.1)/0.9, thus random predictions will have weight 0 and the perfect prediction has weight 1. So if the task has n classes, our method will weight the samples by (pi-1/n)/(1-1/n). There is really no reason why we want to tune these two parameters, as we are weighting the data points by probability.
>
> As suggested by one of the reviewers, we have provided further sensitivity analysis of our method on the confidence threshold. Our default confidence threshold value is set to be 0.8. And we have experiment it with some other confidence values, 0.6, 0.7 and 0.9. The experiment results are on the revised paper. Please check the results in Figure 4 in the Appendix. The results show our method is also robust against confidence threshold value.
>
> As also suggested by one of the reviewers, we have provided further experiment on the more challenging Office-Home dataset. Our method performs the best in all transfer tasks than state-of-the-art UDA methods. The results are in our revised paper (Table 3 in the Appendix), you are welcomed to check that out.
>
> We argue that our work has been severely undervalued by the reviewers. If you think our argument is invalid in some aspects, please indicate. Thanks.
>
>
> [1] Yingwei Pan, Ting Yao, Yehao Li, Yu Wang, Chong-Wah Ngo, and Tao Mei.  Transferrable proto-typical networks for unsupervised domain adaptation. InProceedings of the IEEE Conference on Computer Vision and Pattern Recognition, pp. 2239–2247, 2019.
> [2] Weitao  Wan,  Yuanyi  Zhong,  Tianpeng  Li,  and  Jiansheng  Chen.   Rethinking  feature  distributionfor loss functions in image classification.2018 IEEE/CVF Conference on Computer Vision andPattern Recognition, pp. 9117–9126, 2018.
> [3] Jake Snell, Kevin Swersky, and Richard Zemel. Prototypical networks for few-shot learning. InAdvances in Neural Information Processing Systems, pp. 4077–4087, 2017.
> [4] Mingsheng Long, Yue Cao, Jianmin Wang, and Michael I Jordan.  Learning transferable featureswith deep adaptation networks.arXiv preprint arXiv:1502.02791, 2015.
> [5] Baochen Sun and Kate Saenko.  Deep coral: Correlation alignment for deep domain adaptation.  InEuropean Conference on Computer Vision, pp. 443–450. Springer, 2016.
> [6] Werner  Zellinger,   Thomas  Grubinger,   Edwin  Lughofer,   Thomas  Natschl ̈ager,   and  SusanneSaminger-Platz.   Central moment discrepancy (cmd) for domain-invariant representation learn-ing.arXiv preprint arXiv:1702.08811, 2017.
> [7] Geoffrey  French,  Michal  Mackiewicz,  and  Mark  H.  Fisher.   Self-ensembling  for  visual  domainadaptation. InICLR, 2018.

---

> ### Author Response · Authors · 2019-11-07
> **Response to Reviewer #2 Part 1**
>
> Thanks for reviewing our paper. You rejected our paper based on two reasons: "This paper should be rejected because (1) the novelty of the main idea is marginal, and (2) the performance gain over the baseline methods is also marginal.". We know it is difficult to argue about the novelty part, as different people have different tastes, however we want to have a try.
>
> As a researcher in the area of domain adaptation, you and us all agree on the importance of this area and have read a lot of great works and come across tons of ideas in this area. But in all of these works and ideas, as far as we are concerned, none of them thought about modeling the feature distribution for domain adaptation though it facilitates us to better measure the distribution discrepancy across domains. This is the gap in the area of domain adaptation we are trying to fill with this work. You mentioned "the novelty of the main idea is marginal", so we want to ask which work do you have in mind that generates our idea as a marginal when you claim that? If you have, please provide us the example. Thanks very much.
>
> You mentioned "Pan et al.[1] already proposed the idea of transferring the knowledge from the source to the target using the prototype of each class.", however, we want to clarify again that our work is not about applying prototypical network for domain adaptation, the main idea in our work is to model the feature distribution for domain adaptation, which is a new methodology for domain adaptation. Thus, inspired from Wan et al.'s [2] work, we model the feature distribution as Gaussian Mixture. In the Related Works section, we cite Snell et al.'s [3] paper, showing that learning prototypical network is equivalent to modeling feature distribution as exponential density. This statement shows the only connection between our work and Pan et al.'s. However, the equivalence expressed in the statement is only true for training a model in a single domain. Our work is way different from Pan et al.'s work in the setting of domain adaptation.
> First, we base on different ideas. While Pan et al. propose a novel idea to remold Prototypical Network (PN) for domain adaptation, as stated in their paper, our work is based on the idea that almost all existing domain adaptation methods are minimizing the feature distribution discrepancy for effective knowledge transfer from source domain to target domain, however none of them explicitly models the feature distribution though intuitively it facilitates the measuring of distribution discrepancy, thus minimizing the measurement reduces the discrepancy.
> Second, the two works propose different distribution discrepancy loss functions. While Pan et al. proposes multi-granular distribution discrepancy loss functions at both class-level and sample-level. Our work proposes two novel distribution discrepancy loss function based on probability, one is Gaussian Component Mean Matching and one is Pseudo Distribution Matching. The two distribution discrepancy loss functions work at different aspects and complement each other, where GCMM brings the two distribution closer, PDM shapes the two distribution alike. The two distribution discrepancy loss functions also work at different levels. GCMM reduces domain discrepancy at class level, while PDM reduces domain discrepancy at sample level. We all know that discrepancy loss functions play the central role in a domain adaptation method and devising new domain discrepancy loss functions for domain adaptation is an active research area [4,5,6]. Thus, researchers in the area of domain adaptation would not ignore the two novel discrepancy loss functions we put forward. Furthermore, the idea that modeling the feature distribution enables us to propose new distribution discrepancy loss functions inspires further exploration in this direction to device more novel distribution discrepancy measures.
>
> You further mentioned "It is required to explain why explicit modeling performs better than implicit modeling of prototypes by theory or practice.". As we are exploring the direction of modeling feature distribution for domain adaptation, we do not have much theory to back it up currently. Indeed, modeling the feature distribution as Gaussian Mixture enables us to propose two novel domain discrepancy loss functions. One is Gaussian Component Mean Matching (GCMM) and one is Pseudo Distribution Matching (PDM). For GCMM, Pan et. al. have proposed a similar one, which they called general purpose domain adaptation, but theirs is more complicated and does not inherit a probability interpretation.

---

### Decision · Program_Chairs · 2019-12-19

**Decision:**

Reject

**Comment:**

This paper addresses the problem of unsupervised domain adaptation and proposes explicit modeling of the source and target feature distributions to aid in cross-domain alignment.

The reviewers all recommended rejection of this work. Though they all understood the paper’s position of explicit feature distribution modeling, there was a lack of understanding as to why this explicit modeling should be superior to the common implicit modeling done in related literature. As some reviewers raised concern that the empirical performance of the proposed approach was marginally better than competing methods, this experimental evidence alone was not sufficient justification of the explicit modeling. There was also a secondary concern about whether the two proposed loss functions were simultaneously necessary.

Overall, after reading the reviewers and authors comments, the AC recommends this paper not be accepted.